# *Entamoeba histolytica*: In Silico and In Vitro Oligomerization of EhHSTF5 Enhances Its Binding to the HSE of the *EhPgp5* Gene Promoter

**DOI:** 10.3390/ijms25084218

**Published:** 2024-04-11

**Authors:** Salvador Pérez-Mora, David Guillermo Pérez-Ishiwara, Sandra Viridiana Salgado-Hernández, María Olivia Medel-Flores, César Augusto Reyes-López, Mario Alberto Rodríguez, Virginia Sánchez-Monroy, María del Consuelo Gómez-García

**Affiliations:** 1Laboratorio de Biomedicina Molecular 1, Escuela Nacional de Medicina y Homeopatía, Instituto Politécnico Nacional, Mexico City 07320, Mexico; sperezm1510@alumno.ipn.mx (S.P.-M.); dperez@ipn.mx (D.G.P.-I.); ssalgadoh1600@alumno.ipn.mx (S.V.S.-H.); molivof@ipn.mx (M.O.M.-F.); 2Laboratorio de Bioquímica Estructural, Escuela Nacional de Medicina y Homeopatía, Instituto Politécnico Nacional, Mexico City 07320, Mexico; careyes@ipn.mx; 3Departamento de Infectómica y Patogénesis Molecular, Centro de Investigación y de Estudios Avanzados del IPN (Cinvestav), Mexico City 07360, Mexico; marodri@cinvestav.mx; 4Sección de Posgrado e Investigación, Escuela Superior de Medicina, Instituto Politécnico Nacional, Mexico City 11340, Mexico; vsanchezm@ipn.mx

**Keywords:** *Entamoeba histolytica*, rEhHSTF5, oligomerization, monomer, dimer, trimer, HSE_*EhPgp5*, EhDBD5, molecular docking

## Abstract

Throughout its lifecycle, *Entamoeba histolytica* encounters a variety of stressful conditions. This parasite possesses Heat Shock Response Elements (HSEs) which are crucial for regulating the expression of various genes, aiding in its adaptation and survival. These HSEs are regulated by Heat Shock Transcription Factors (EhHSTFs). Our research has identified seven such factors in the parasite, designated as EhHSTF1 through to EhHSTF7. Significantly, under heat shock conditions and in the presence of the antiamoebic compound emetine, EhHSTF5, EhHSTF6, and EhHSTF7 show overexpression, highlighting their essential role in gene response to these stressors. Currently, only EhHSTF7 has been confirmed to recognize the HSE as a promoter of the *EhPgp5* gene (HSE_*EhPgp5*), leaving the binding potential of the other EhHSTFs to HSEs yet to be explored. Consequently, our study aimed to examine, both in vitro and in silico, the oligomerization, and binding capabilities of the recombinant EhHSTF5 protein (rEhHSTF5) to HSE_*EhPgp5*. The in vitro results indicate that the oligomerization of rEhHSTF5 is concentration-dependent, with its dimeric conformation showing a higher affinity for HSE_*EhPgp5* than its monomeric state. In silico analysis suggests that the alpha 3 α-helix (α3-helix) of the DNA-binding domain (DBD5) of EhHSTF5 is crucial in binding to the major groove of HSE, primarily through hydrogen bonding and salt-bridge interactions. In summary, our results highlight the importance of oligomerization in enhancing the affinity of rEhHSTF5 for HSE_*EhPgp5* and demonstrate its ability to specifically recognize structural motifs within HSE_*EhPgp5*. These insights significantly contribute to our understanding of one of the potential molecular mechanisms employed by this parasite to efficiently respond to various stressors, thereby enabling successful adaptation and survival within its host environment.

## 1. Introduction

*Entamoeba histolytica* (*E. histolytica*), the causative agent of human amoebiasis, primarily infects the intestinal tract. However, it can also spread to other organs, giving rise to severe complications such as hepatic abscesses, pneumonia, purulent pericarditis, and in rare cases, cerebral amebiasis [1,2]. It is estimated that this pathogen is responsible for approximately 50 million infections worldwide annually, with a resultant morbidity that exceeds 100,000 deaths each year [3].

Throughout the course of host infection, the parasite encounters a myriad of stressful conditions, including acidic pH levels, the presence of noxious agents, engagement with the host’s immune defenses, drug exposure, heavy metal toxicity, hypoxia, nutrient scarcity, and thermal fluctuations, among others. Despite these hostile environments, the parasite demonstrates remarkable adaptive capabilities, enabling it to persist and induce amoebiasis [4,5,6,7]. Therefore, it is important to understand the molecular mechanisms that trophozoites employ to overcome stressful situations and ensure their survival.

Among the wide variety of proteins present in organisms that can modulate diverse processes are the Heat Shock Transcription Factors (HSTFs).

There is considerable variation in the number of HSTFs identified across different species along the evolutionary scale. The number of genes encoding HSTFs ranges widely, from a single gene in some organisms to many in others. For instance, organisms such as *Caenorhabditis elegans* (*C. elegans*, nematode worm), *Saccharomyces cerevisiae* (*S. cerevisiae*, baker’s yeast), and *Drosophila melanogaster* (*Drosophila melanogaster*, fruit fly), each have just one HSTF gene. In contrast, other organisms have been found to possess a higher number of these genes: *Homo sapiens* (*H. sapiens*, humans) have six, *Arabidopsis thaliana* (*A. thaliana*, thale cress) has twenty-one, and *Solanum lycopersicum* (*S. lycopersicum*, tomato) has twenty-four. Remarkably, *Glycine max* (*G. max*, soybean) stands out with fifty-three HSTF genes [8,9]. Interestingly, by the year 2016, 848 coding sequences for HSTFs from 33 plant species had been registered in the Heatster database (database version 1.0, available at: https://applbio.biologie.uni-frankfurt.de/hsf/heatster/, accessed on 5 April 2024), highlighting the important role of HSTFs in organisms.

HSTFs have a structurally conserved DNA-Binding Domain (DBD), featuring three alpha helices and four folded β-sheets, an Oligomerization Domain (OD) with a coiled-coil motif for dimer and trimer formation, and a Carboxyl-terminal Transactivation Domain (CTD). The transcriptional activity of HSTFs is intricately controlled by both intra- and inter-molecular interactions, as well as by a variety of Post-Translational Modifications (PTMs) [10,11].

Under normal conditions, HSTFs are inactive and present as monomers in the cytoplasm. However, under certain conditions, these transcription factors are activated through various PTMs, including phosphorylation. This modification promotes their oligomerization into dimers and trimers, which subsequently translocate to the nucleus. Once there, HSTF trimers, through their DBDs, recognize and bind specifically to HSEs as the promoters of target genes, thereby modulating their expression [10,12]. Given the critical role of HSTFs, their correct activation is essential, allowing them to respond to cellular stress. A failure in this process leaves HSTFs inactive, preventing their protective function. This can result in cellular damage, potentially leading to diseases or cellular death [13]. Upon oligomerization, HSTFs transform into a more stable structure, enhancing their affinity towards HSEs and facilitating the precise regulation of crucial genes for maintaining cellular homeostasis. Thus, oligomerization becomes a fundamental process for the effective activation and performance of the protective function of HSTFs, underscoring the importance of researching this process [13,14].

HSTFs are pivotal in ensuring cellular survival under a broad spectrum of environmental and physiological stresses, playing a critical role in safeguarding cellular integrity [14,15]. Their principal function involves the activation of heat shock gene expression, especially those genes encoding Heat Shock Proteins (HSPs). Acting as molecular chaperones, HSPs are crucial in the correct folding of proteins, preventing the formation of detrimental protein aggregates and facilitating the repair or degradation of damaged proteins. This process is essential for maintaining both protein stability and cellular functionality [16]. Beyond their primary role, HSTFs also significantly contribute to several other critical cellular processes, including development, differentiation, senescence, and apoptosis [17]. Additionally, HSTFs are essential in the pathogenesis and progression of a variety of diseases due to their role in regulating HSPs and associated cellular processes. In neurodegenerative diseases, such as Alzheimer’s [18] and Parkinson’s [19], HSTFs can help mitigate the accumulation of misfolded proteins. In cancer, they promote the survival and resistance of tumor cells to treatments [20]. In the context of cardiovascular diseases, they protect the heart from stress damage [21]. HSTFs also play a role in inflammatory diseases [22], such as rheumatoid arthritis [23], affecting their severity and progression.

Research into HSTFs in parasites remains scant. To date, only a few studies, including a notable one by Gelmedin et al. [24] have explored this area. They demonstrated that HSTF1 is essential for the development from larva to the adult stage in the hookworm *Ancylostoma caninum*, providing protection against thermal shock during this process and throughout the infection. Another study highlighting the importance of HSTFs in parasites is by Levyet al. [25] This research explored the expression of various forms of HSTFs throughout the life cycle of the parasitic helminth *Schistosoma mansoni*. It was discovered that two or three possible conformations of HSTFs can recognize an HSE both at the schistosomula stage and in adult worms, suggesting that HSTFs are essential for adapting to different developmental stages and responding effectively to heat stress conditions during the parasite life cycle.

Particularly, according to the AmoebaDB database (https://amoebadb.org/amoeba/app, accessed on 15 December 2023) [26], the *E. histolytica* HSTFs (Heat Shock Transcription Factors), referred to as EhHSTFs, until now, were considered the second largest family after those in plants. This family consists of 7 members, identified as *Ehhstf1* (EHI_008230), *Ehhstf2* (EHI_049510), *Ehhstf3* (EHI_087630), *Ehhstf4* (EHI_142120), *Ehhstf5* (EHI_137000), *Ehhstf6* (EHI_008660), and *Ehhstf7* (EHI_200020).

Currently, only the EhHSTF7 protein has been shown to have the ability to oligomerize and recognize the HSE in the promoter region of the *EhPgp5* gene (HSE_*EhPgp5*), both in vitro and in vivo. This gene regulation mechanism, exerted by the HSTFs on the HSE_*EhPgp5*, is crucial, as this gene is responsible for encoding a transporter protein known for conferring resistance to multiple drugs, which allows the amoeba to defend against various pharmaceutical compounds [27]. Additionally, these factors have the capacity to regulate HSPs, thereby enhancing cellular survival in response to various types of stress. This regulation can extend to other vital processes in the parasite, including invasion, virulence, interaction with the host immune system, encystation, the transition from cyst to trophozoite, and the regulation of genes linked to metabolism as detailed by Dorantes et al. [28]. This versatility underscores the importance of HSTFs in the life cycle of the parasite and its ability to adapt to diverse environments and challenges. Therefore, it is critical to investigate the oligomerization (activation) of EhHSTFs and their affinity for HSEs in *E. histolytica* since these interactions have not yet been thoroughly explored, except for EhHSTF7 [27]. Therefore, the objective of this study was to investigate both the potential oligomerization and the DNA-binding affinity of the recombinant EhHSTF5 protein (rEhHSTF5) to the HSE_*EhPgp5*. In this study, we demonstrate both in vitro and in silico that EhHSTF5 from *E. histolytic* is capable of oligomerizing into dimers and trimers. Additionally, oligomerization is crucial for increasing the affinity for HSE_*Pgp5*, and very likely, by recognizing HSEs, it may also be capable of regulating the expression of a wide variety of genes harboring HSEs in their promoters. This opens a new perspective on this transcription factor, positioning it as a potential molecular target against amebiasis.

## 2. Results

### 2.1. The Oligomeric Conformations of rEhHSTF5 Protein Are Recognized by α6xHis and αEhHSTF5 Antibodies

The Open Reading Frame (ORF) of the *Ehhstf5* gene was successfully inserted into the pET-28a(+) plasmid and then confirmed through DNA sequencing and enzymatic restriction (Appendix A). We then proceeded to induce the expression of the recombinant EhHSTF5 protein (rEhHSTF5) in *Escherichia coli* (*E. coli*) C41, as illustrated in Figure 1a. Induction assays revealed the overexpression of different proteins with approximate molecular weights of 38, 47, 76, and 114 kDa at different induction times using isopropyl-β-D-thiogalactopyranoside (IPTG). Western blot (WB) assays with the α6xHis antibody recognized the 38 and 76 kDa proteins, while the αEhHSTF5 antibody detected all four proteins (Figure 1b).

These results indicate that the 38, 76, and 114 kDa proteins probably represent different oligomerization states of the rEhHSTF5 protein, corresponding to the monomer, dimer, and trimer states, respectively. The 47 kDa band may represent the protein in a transition state toward oligomerization. The monomeric conformation of the recombinant EhHSTF5 protein (mrEhHSTF5) was consistently observed at all induction times evaluated. It peaked at 3 h after induction and gradually decreased. In contrast, the detection of the dimeric conformation of rEhHSTF5 (drEhHSTF5) was minimal at 3 h but increased markedly at 6, 9, and 24 h, whereas the trimeric conformation of rEhHSTF5 (trEhHSTF5) was uniquely identified by the αEhHSTF5 antibody after 24 h of induction with IPTG. This result suggests that increasing the protein induction time leads to a higher concentration in the cellular environment, consequently inducing oligomerization.

### 2.2. Purification of the rEhHSTF5 Protein in Its Monomeric and Dimeric Conformations

The nickel affinity chromatography was employed to purify the rEhHSTF5 protein. When the total protein extract was processed through the nickel column, a significant decrease in the concentration of mrEhHSTF5 was observed in the column filtrate, suggesting a strong affinity for the column. Conversely, the dimeric and trimeric conformation of rEhHSTF5 showed only a slight reduction in the filtrate, indicating either a lower affinity for the column or possible saturation of the column by mrEhHSTF5, preventing their retention. During the elution process, a total of 60 fractions were collected. The dimeric conformation primarily eluted in fractions 7 to 12, while the monomeric conformation was found in fractions 24 to 26. However, trEhHSTF5 was not detected in these fractions (Figure 2a–d). To verify the identities of the purified proteins corresponding to the monomeric and dimeric conformations of rEhHSTF5, Western blot assays were conducted using α6xHis and αEhHSTF5 antibodies. These assays successfully demonstrated the immunodetection of both protein conformations, as depicted in Figure 2e,f.

### 2.3. The rEhHSTF5 Protein Undergoes Oligomerization

Crosslinking assays were performed using the protein predominantly in its monomeric conformation (fraction 25) to demonstrate the oligomerization capability of mrEhHSTF5. In the absence of glutaraldehyde, the mrEhHSTF5 protein was the most frequently observed, while drEhHSTF5 and trEhHSTF5 were detected with less intensity. However, with the addition of glutaraldehyde, a concentration-dependent decrease in the intensity of mrEhHSTF5 was noted. At higher concentrations of glutaraldehyde, drEhHSTF5 disappeared, and the intensity of trEhHSTF5 increased (Figure 3). The formation of a 47 kDa protein was observed only in samples with glutaraldehyde at various concentrations, supporting the hypothesis that rEhHSTF5 undergoes transitional states that enable it to oligomerize into dimeric and trimeric conformations, with approximate molecular weights of 76 and 114 kDa, respectively. These molecular weights are consistent with those of proteins detected by α6xHis and αEhHSTF5 antibodies in IPTG-induced bacteria.

### 2.4. The EhPgp5 Gene Promoter HSE Is Recognized by the rEhHSTF5 Protein

Electrophoretic Mobility Shift Assay (EMSA) experiments were conducted to assess whether the rEhHSTF5 can recognize the HSE_*EhPgp5*. For this, biotinylated synthetic double-stranded oligonucleotides containing HSE_*EhPgp5* were incubated with the mrEhHSTF5 (fraction 25) and the drEhHSTF5 (fraction 10).

The experiments revealed an interesting outcome: drEhHSTF5, but not mrEhHSTF5, formed a DNA–protein complex under the same experimental conditions. To determine the specificity of the drEhHSTF5-HSE interaction, both specific and nonspecific competitors were used. The formation of the drEhHSTF5-HSE complex was reduced with the specific competitor, whereas it remained evident with Poly (dI-dC), as shown in Figure 4a. This indicates that the dimeric conformation of rEhHSTF5 has a higher and more specific affinity for the HSE sequence compared to its monomeric state.

Additionally, the use of the αEhHSTF5 antibody resulted in the formation of αEhHSTF5-drEhHSTF5-HSE_*EhPgp5* supercomplexes, which migrated more slowly compared to the drEhHSTF5-HSE_*EhPgp5* complex in the absence of the αEhHSTF5 antibody. Interestingly, by pre-incubating mrEhHSTF5 with the specific antibody αEhHSTF5 and, subsequently with biotinylated HSE, the formation of the supercomplex αEhHSTF5-mrEhHSTF5-HSE_*EhPgp5* was observed. It is noteworthy that no changes in the mobility of the complex (supershifts) were detected when nonspecific antibodies were used (Figure 4b). Furthermore, increasing the concentration of the drEhHSTF5 protein led to an enhancement in the formation of both αEhHSTF5-drEhHSTF5-HSE_*EhPgp5* and drEhHSTF5-HSE_*EhPgp5* complexes, as illustrated in Figure 4c. These results confirm the specific binding capability of the drEhHSTF5 factor to HSE_*EhPgp5*.

### 2.5. The DNA-Binding Domain of EhHSTF5 Has a Highly Conserved Three-Dimensional Structure

To predict the behavior and binding mechanisms of the rEhHSTF5 protein in its monomeric and dimeric conformations to HSE_*EhPgp5*, three-dimensional (3D) models were constructed for HSE_*EhPgp5* (Figure 5a), and for the DBD of the monomeric (mEhDBD5, Figure 5b) and dimeric (dEhDBD5, Figure 5c) conformations of EhHSTF5.

Upon analyzing the secondary structure using the PDBsum server, we found that the 3D models for both mEhDBD5 and dEhDBD5 exhibit the typical and highly conserved structure of HSTFs, featuring three α-helices and four β-strands. Additionally, the structures included seven β-turns, one γ-turn, and two β-hairpins. To further understand the potential stability of EhDBD5, we predicted stability using the I-TASSER server. Interestingly, the analysis of the primary sequence of mEhDBD5 showed that the three α-helices and the four β-strands had normalized B-factor values of 0 or less, suggesting these secondary structures are rigid and stable. In contrast, regions of the amino acid sequence without well-defined secondary structures, such as coils, displayed B-factor values of 0 or greater, indicating that these areas are less stable due to increased flexibility or mobility, which could influence the behavior of the protein and its binding properties.

### 2.6. The 3D Models of mEhDBD5 and dEhDBD5 Exhibit Satisfactory Structural Quality

The 3D models of the mEhDBD5 and dEhDBD5 demonstrated excellent stereochemical quality. This quality was confirmed by analyzing the Phi/Psi torsion angles of the peptide bonds through Ramachandran analysis. In this analysis, all amino acids were found in favored regions (100%), with none in disallowed regions (0%). Additionally, on the ProSA-web server, the Z-scores for mEhDBD5 and dEhDBD5 were −6.09 and −6.67, respectively. These scores fall within the typical range for similar structures determined through crystallography, indicating the reliability of the models. The energy-based evaluation of the models further reinforced their high quality, with values being ≤ 0. Further validation was conducted using ERRAT, which yielded an overall quality factor of 100 for mEhDBD5 and 87.08 for dEhDBD5. This suggests a high level of accuracy and reliability in the model structures. The 3D VERIFY analysis also supported these findings, indicating that 100% of the residues in mEhDBD5 and 84.59% in dEhDBD5 had a 3D−1D score of ≥0.1 (Figure 6).

In addition, a comprehensive structural quality assessment of the mEhDBD5 and dEhDBD5 models was conducted, focusing on key physicochemical and geometric characteristics. This evaluation included several quality indices, including Fractional Accessible Surface Area (FASA), Fractional Residual Volume (FRV), Stereo/Packed Quality Index (SPQI), and 3D Profile Quality Index (3DPQI). These indices were crucial in providing a detailed analysis of various aspects of the models, such as surface exposure, internal packing, stereochemical configuration, and overall 3D conformation. It is important to highlight that both mEhDBD5 and dEhDBD5 obtained scores that are within the range were suggested by the VADAR server, indicating an excellent structural quality of the 3D models obtained (Appendix A).

### 2.7. dEhDBD5 Exhibits Higher Affinity for the HSE of the EhPgp5 Gene Promoter Than mEhDBD5

Through blind molecular docking, both mEhDBD5 and dEhDBD5 successfully recognized HSE_*EhPgp5* (Figure 7). The dEhDBD5 exhibited a higher binding energy compared to mEhDBD5, as indicated by more negative values. Docking scores for the mEhDBD5-HSE_*EhPgp5* and dEhDBD5-HSE_*EhPgp5* complexes were −203.29 and −305.79, respectively, with corresponding confidence scores of 0.74 and 0.95. The first dockings were selected due to the quality of their parameters.

Intermolecular analysis indicated that both mEhDBD5 and dEhDBD5 primarily bind to the major groove of HSE_*EhPgp5* through the α3-helix. On the sense strand of HSE_*EhPgp5*, mEhDBD5 formed five hydrogen bond interactions and one salt bridge, while on the complementary strand, it engaged in seven hydrogen bond interactions and one salt bridge. In contrast, dEhDBD5 showed more extensive interactions: eight hydrogen bonds and four salt bridges on the sense strand, and nine hydrogen bonds and six salt bridges on the complementary strand (Appendix A).

In the mEhDBD5-HSE_*EhPgp5* complex, the amino acids Arg 55 and Asn 58 interact with the adenine of the “GA**A**” motif in the sense strand, while Lys 92 forms a salt bridge with the phosphate group located between two adenines in “GA**AA**”. In the complementary strand, Ile 1 and Tyr 60 interact with an adenine of the “**A**AG” motif. In contrast, in the dEhDBD5-HSE_*EhPgp5* complex, we identified that monomer 1 (in red) interacted in the sense strand with Gln 56 with the adenine “G**A**A”, Ser 48, Asn 49, Ser 52 with “GA**A**”, and His 47 formed a salt bridge with the phosphate group located between the adenines in “G**AA**”. Monomer 2 (in yellow) interacted in the complementary strand with Glu 66 with the cytosine of the “**C**TT”motif, Arg 55 with “**A**AG”, Ser 52 with “A**A**G”, and Gln 56 with “AA**G**”. We identified three salt bridges, Lys 64 between two adenines in “GA**AA**” in the sense strand, and in the complementary strand, Lys 64 between two adenines of the “**AA**G” motif, and His 47 with the phosphate group located between the adenine and guanine in “A**AG**” in the complementary strand (Figure 7a–c).

Interestingly, in the crystallized complexes of KlHSTF (PDB ID: 3HTS), HsHSTF1 (PDB ID: 5D5X), HsHSTF2 (PDB ID: 5D8L), and in the in silico predictions for the EhHSTF7-HSE_*Ehpgp5* complex [27], the amino acids Arg, Ser, and Gln from α3-helix play a crucial role in mediating interactions with the “GAA” and “TTC” motifs (Figure 8c). Complementarily, upon aligning and generating a logo of the amino acids present in the α2-helix and α3-helix of EhDBD5 with 28 sequences of HSTF factors from diverse species, including the EhHSTF family, we found that α3-helix exhibits a high degree of conservation. Notably, amino acids such as Ser, Phe, Arg, Gln, Leu, Asn, Tyr, and Gly are prominently conserved (Figure 7d).

### 2.8. The Intermolecular Interaction Mechanism of the dEhDBD5-HSE Complex Is Highly Conserved

To provide a detailed comparison between the dEhDBD5-HSE_*EhPgp5* complex and similar crystallographic complexes, the RaptorX server was utilized. This server offers two key parameters for comparison: the Structural Similarity Percentage (TMscore) and the Root Mean Square Deviation (RMSD). A TMscore value of 60% or higher indicates a significant similarity in conformation and structure between compared complexes, while the RMSD measures the proximity of superimposed structures in three-dimensional space.

Our in silico analysis revealed that the dEhDBD5-HSE_*EhPgp5* complex displayed TMscore values of 81% or greater and RMSDs of 1.4 or lower when compared with crystallographic complexes of *K. lactis*, *S. cerevisiae*-*C. thermophilum* (chimera), *H. sapiens*-*C. thermophilum* (chimera), synthetic *H. sapiens*, and native *H. sapiens*, as shown in Figure 8b–f. Upon overlaying all these structures, an average TMscore of 89.8% and an RMSD of 1.17 were observed. This suggests a high degree of structural conservation and indicates functional conservation, as evidenced by the similar spatial conformation when binding to HSEs (Figure 8g). A particularly noteworthy finding emerged when only the α2-helix and α3-helix of the DBDs were superimposed, resulting in an average TMscore exceeding 85.5% (Figure 8h). Additionally, upon comparing the primary amino acid sequences of these 3D structures, the α2-helix and α3-helix of EhDBD5 demonstrated identity and homology percentages above 73% in comparison to the evaluated crystallographic structures. This further emphasizes the significant structural and functional conservation of these domains, particularly in their roles related to DNA binding and transcriptional regulation.

An analysis of helix α2 and helix α3 in EhDBD5 at the primary sequence level showed that their identity and homology was 73% and 76% or higher, respectively, compared to the same helices in the five evaluated crystallographic structures. Additionally, at the 3D structural level, these α-helices in EhDBD5 exhibited TM scores of 73.5 or higher, suggesting a conserved folding pattern. This result also underscores the high conservation of amino acids such as Asn, Ser, Phe, Val, Arg, Gln, Leu, and Tyr in the α3-helix (Figure 8i).

## 3. Discussion

In this study, using WB assays, we demonstrated that the rEhHSTF5 protein undergoes oligomerization, forming dimers and trimers in protein extracts from bacteria harboring the recombinant protein. The induction with IPTG for 3 h enhances the formation of mrEhHSTF5, while induction at 24 h favors the formation of drEhHSTF5 and trEhHSTF5, suggesting that oligomerization is concentration-dependent. Protein oligomerization has been documented in HSTFs from various species. HSTF1 [15], HSTF2 [29], and HSTF4 [30] from *H. sapiens*; HSTF1 from *A. thaliana* [31], HSTF1 from *D. melanogaster* [32], and EhHSTF7 from *E. histolytica* [27], among others. These transcription factors tend to oligomerize in cellular environments with elevated protein concentrations, while remaining as monomers at lower concentrations.

On the other hand, the 47 kDa protein expressed during IPTG induction, which was recognized by the αEhHSTF5 antibody, suggesting that the 47 kDa protein may represent a stable conformation. This conformation could be an important transient state for dimer and trimer formation, which allows for the interaction and coordinated assembly of the subunits, maintaining the stability of the oligomeric structure as it occurs in different proteins [33], including Superoxide Dismutase 1 (SOD1) [34], or Heat Shock Protein 90 (HSP90) [35], among others.

Additionally, through nickel affinity chromatography, we successfully purified the rEhHSTF5 protein in both its monomeric and dimeric conformations. The monomeric protein was used for cross-linking assays with glutaraldehyde. Interestingly, we found that increasing the concentration of glutaraldehyde led to the oligomerization of rEhHSTF5, forming dimers and trimers with estimated weights of 38 and 114 kDa, respectively, which are consistent with those detected by WB in total extracts. Another notable observation was that, in the presence of glutaraldehyde, we again identified the formation of a 47 kDa protein, further supporting the hypothesis that this may represent a stable conformation of the protein during the transition to oligomerization.

By WB assays, we found that the oligomerization of the mrEhHSTF5 protein can hide the polyhistidine tag, which makes it difficult to detect using the α6xHis antibody in induced bacteria extracts. Furthermore, after conducting nickel-affinity chromatography, we were unsuccessful in purifying the 114 kDa protein. This reinforces the idea that the polyhistidine region is hidden, making it difficult to bind to the nickel column. This observation is consistent with the behavior of the rEhHSTF7 protein reported by Bello et al. [27], whereas, under similar experimental conditions, they were both unable to detect the trimeric conformation of the protein using the α6xHis antibody nor to purify it through nickel affinity purification. Moreover, it has been previously demonstrated that the oligomerization of recombinant proteins can alter the accessibility of polyhistidine tags, potentially affecting their binding capacity to affinity matrices [36,37,38].

HSTFs are recognized for their role in heat stress response and their ability to bind to HSEs to regulate gene expression [39,40,41]. The oligomerization and stability of these factors are of vital importance for their proper function, as their ability to bind to HSE and activate transcription depends on the formation of stable oligomeric structures. HSTF1 and HSTF2 from *H. sapiens* form homotrimers or heterotrimers with other HSTFs. This oligomer formation increases their structural stability and their affinity for HSE, resulting in an increased ability to regulate gene expression. Therefore, oligomerization is a critical factor in achieving the necessary stability to recognize HSE [42,43,44]. This fact is in concordance with our EMSA assays demonstrating that the dimeric conformation of the rEhHSTF5 protein exhibits a higher affinity for binding to the HSE_*EhPgp5*. In contrast, the monomeric conformation under the same experimental conditions did not bind with the HSE_*EhPgp5*, confirming the necessity of oligomerization to achieve greater stability and, consequently, higher affinity to the HSE_*EhPgp5*. Supershift assays validated the EhHSTF5-HSE interaction, forming the αEhHSTF5-drEhHSTF5-HSE supercomplex.

The binding of mrEhHSTF5 to the HSE_*EhPgp5* was not detectable by EMSA assays. However, when the protein was preincubated with the αEhHSTF5 antibody, the formation of a highly specific supercomplex was observed. These results suggest that the antibody binding to mrEhHSTF5 induces a conformational change that exposes the DBD and, consequently, enables binding to the HSE_*EhPgp5*. A similar finding was previously reported by Zimarino et al. [45], who found that the binding activity to the HSE by the inactive form of the HSTF from *D. melanogaster* could be induced in vitro by the addition of a polyclonal antibody against the purified factor, proposing that the binding of an anti-HSTF antibody led to a conformational change, exposing its DBD to activate its transcriptional capability. Moreover, the studies performed with HsHSTF1 showed that its DBD is hidden when it is in a monomeric conformation, but is exposed when it is oligomerized, which allows it to bind more easily to HSE [17,40,42]. Similarly, we suggest the possibility that EhDBD5 is hidden in the monomeric conformation of rEhHSTF5 and is exposed upon oligomerization or antibody binding, which would more effectively facilitate interaction with HSE_*EhPgp5*.

In silico analysis is an important tool for understanding protein activity and molecular mechanisms involved [46].

In our study, utilizing the SWISS-Model, we successfully constructed 3D models of EhDBD5 in both its monomeric and homomeric conformations. These models were validated through Ramachandran plot analysis by PDBsum, alongside assessments with ProSA-web, ERRAT, VERIFY 3D, and VADAR. These evaluations demonstrated excellent structural quality of the models. Simultaneously, our analysis using the PDBsum server revealed that the 3D models of mEhDBD5 and dEhDBD5 kept the typical and highly conserved structure of HSTFs, which includes three α-helices and four β-strands. These models also included seven β-turns, one γ-turn, and two β-hairpins. Additionally, when assessing the potential stability of EhDBD5 using the I-TASSER server, we discovered that the α-helices and β-strands are particularly stable, with normalized B-factor values of zero or less, highlighting their structural rigidity compared to other less defined regions of the amino acid sequence.

The 3D structural representations of HSTF-DBDs, like the obtained EhDBD5, have been documented. These include the 3D crystallographic structure of the DBD of HSTF1 from *H. sapiens*, as reported by Neudegger et al. [47]; the DBD of HSTF1 from *K. lactis*, reported by Littlefield and Nelson et al. [48]; and the structures of the DBDs of HSTF1, HSTF2, and HSTF4 from *H. sapiens*, published by Xiao et al. [49]. Additionally, the structure of EhDBD7 from EhHSTF7 of *E. histolytica* was detailed by Bello et al. [27].

Through our in silico blind molecular docking studies using the HDOCK server, we identified that mEhDBD5 and dEhDBD5 exhibit binding to HSE_*EhPgp5*. Notably, dEhDBD5 shows significantly higher binding affinity (−305.79) compared to mEhDBD5 (−203.29).

We previously performed molecular docking under the same conditions as in this study, focusing on the trimeric conformation (tEhDBD5) of the EhHSTF5 protein, and discovered its binding to the same HSE with a Docking score of −459 [28]. This clearly indicates that oligomerization of EhHSTF5 into dimers and trimers progressively improves its stability and affinity towards HSE_*EhPgp5*. This behavior has also been demonstrated in other factors such as KlHSTF [48], HsHSTF1 [47], and HsHSTF2 [50], among others.

Using the PyMol software and the PLIP server, we identified that α3-helix plays a crucial role in the interaction of the DBD domain with the major groove of the HSE, especially in the structural motifs “GAA” and “TTC”, through hydrogen bonds and salt bridges. These results are in agreement with various crystallized complexes (HSTF-HSE) where it has been observed that the α3-helix is responsible for direct contact with the major groove of the HSE. In the KlHSTF-HSE complex, for example, amino acids involved in recognizing the structural motifs 5′-TTC-3′, 3′-AAG-5′, and 3′-CTT-5′ are Arg 250, Ser 247, His 242, Gln 252, and Lys 259 [48]. In the HsHSTF1-HSE crystallographic complex, it has been observed that Arg 71, Ser 68, and Gln 72 recognize the structural motifs 5′-GAA-3′ and 3′-CTT-5′ [47], while in the HsHSTF2-HSE complex, Arg 109, Arg 63, Ser 60, and Lys 72 participate in recognizing the motifs 5′-TTC-3′, 5′-GAA-3′, and 3′-CTT-5′ [50].

Notably, during the analysis of sequences comprising the α3-helix, including EhDBD5, a high conservation of amino acids such as Ser, Arg, Gln, Asn, Tyr, and Gly were observed in several species, suggesting their significant relevance in HSE binding. In fact, it has been reported that the mutation of amino acids Ser 247, Arg 250, and Asn 253 in α3-helix of *K. lactis* leads to a lethal phenotype, further underlining the importance of these amino acids in HSTFs [48].

An important aspect in the interaction between HSTFs and HSEs is that certain amino acids can interact with the phosphate group of the HSE backbone through salt bridges. This is the case for amino acids Ser 68, Arg 71, and Gln 72 in the HsHSTF1-HSE complex, and Lys 72, Asn 57, Ser 60, and Lys 53 in the HsHSTF2-HSE complex [47]. This interaction enhances the stability of the DBD, facilitating the correct binding of α3-helix to HSE. In our study, we observed a similar interaction mechanism mediated by salt bridges. We identified the presence of two salt bridges in the mEhDBD5_HSE_*EhPgp5* complex, and ten in the dEhDBD5_HSE_*EhPgp5* complex. Hydrogen bonds are also crucial for the stability of the complexes. Interestingly, we observed 12 intermolecular interactions in mEhDBD5_HSE_*EhPgp5* and 17 in dEhDBD5_HSE_*EhPgp5* (Appendix A). This suggests that the dimeric conformation, through forming a higher number of salt bridges and hydrogen bonds during its binding to HSE, might be promoting greater stability compared to the monomeric conformation.

Consistently, upon evaluating the physicochemical properties using the BIOVIA Discovery Studio software, we observed that the dEhDBD5-HSE_*EhPgp5* complex, compared to the mEhDBD5-HSE_*EhPgp5* complex, exhibits more aromatic interactions, indicating a higher affinity at the binding interface. It also displays a stronger electric charge, suggesting more robust ionic interactions. Additionally, the complex is more hydrophobic, which enhances its structural stability. Notably, the ionization capacity implies diverse electrostatic interactions. The increased exposure to solvent suggests a more intricate structure and higher solubility. These features indicate a complex network of intramolecular and intermolecular interactions in the dimeric complex, potentially contributing to its greater stability compared to the monomeric conformation, as has been documented in numerous studies for different proteins [51,52,53,54].

A study that highlights a similar functional and structural behavior to rEhHSTF5 is the report by Bello et al. [27] which revealed that the rEhHSTF7 factor can oligomerize in a concentration-dependent manner within total extracts of C41 bacteria, and this oligomerization mechanism was verified by αEhHSTF7 and α6xHis antibodies. In addition, the successful purification of the protein in its oligomeric states has been achieved through nickel affinity chromatography. Analogously to the current study, EMSA and supershift assays have demonstrated that the oligomeric form of *E. histolytica* rEhHSTF7 can recognize the HSE_*EhPgp5* gene. It is interesting to observe that through in silico tools, it was confirmed that the α3-helix also recognizes the 3′-AAG-5′ motif, primarily through the amino acids Arg 72 and Ser 69. Moreover, it has been shown that the expression of the *EhPgp5* gene decreases when a specific siRNA is used to knockdown the expression of the *Ehhstf7* gene in *E. histolytica* trophozoites growth with 8 µM of Emetine, underlining the significance of transcriptional modulation by this factor under stress conditions. It is worth highlighting that the EhPGP5 protein of *E. histolytica* is recognized as a multidrug resistance protein, acting as a pump to expel drugs, and enhancing amoebic survival when exposed to amoebicidal drugs. In this work, we observed that oligomeric rEhHSTF5 can recognize the HSE_*EhPgp5* gene similarly to rEhHSTF7, suggesting that both factors could collaborate in regulating the gene transcription of *EhPgp5* and, consequently, help the survival of the amoeba by responding to stressors present in its environment.

On the other hand, the overlay of 3D protein structures through various in silico analyses has been crucial in identifying similarities and conservation of domains, structures, and binding to their consensus DNA sequence, even strongly suggesting conservation in function. Such is the case with the work conducted by Feng et al. [40], who overlaid two crystallographic structures of DBDs analogous to the HSTF1 and HSTF2 trimers from *H. sapiens* bound to an HSE, concluding that both factors are highly conserved both structurally and functionally when binding to the HSE in a similar manner in 3D space. The analysis of overlapping 3D structures has also been applied to proteins other than HSTFs. For instance, in the work of Park et al. [55], the RabGAP domains of TBC1D1 and TBC1D4 are overlaid, concluding that these structures are highly conserved structurally, with 86% similarity, suggesting possible functional conservation. Similarly, the study by Sato et al. [56] involves the overlay of various proteins from the ribokinase family with TK2285, demonstrating that they are highly conserved at the structural level.

Based on these studies, we opted to use the dEhDBD5 complex bound to HSE_*EhPgp5* for comparison with various crystallized structures deposited in the PDB. Interestingly, when we overlaid our complex with crystallographic complexes (DBDs-HSE) from *K. lactis*, a *S. cerevisiae-C. thermophilum* chimera, a *H. sapiens-C. thermophilum* chimera, synthetic *H. sapiens*, and native *H. sapiens*, TMscore values of 81% or higher and RMSDs of 1.4 Å or lower were observed. This suggests that the spatial configuration of our docking closely aligns with the crystallographic structures, indicating a similarity in structural folding and binding to the HSE. This observation underscores the highly conserved mechanism of HSTFs when interacting with HSEs, evident not only for the EhHSTF5 and EhHSTF7 factors of *E. histolytica* but also across various HSTFs from different species.

It is important to emphasize the capability of the rEhHSTF5 factor to recognize the structural motifs “GAA” and “TTC” within the HSE, which expands the potential for recognition not only of the HSE_*EhPgp5* gene [27,57], but also, under specific conditions, the HSEs present in other promoters regulated by this element, such as the seven HSEs identified in *EhrabB* [58], three in *Ehhsp100* [59], and four in the *Ehmlbp* gene promoter [60]. These proteins are of great significance for the parasite, as EhRabB plays a pivotal role in cytoskeleton regulation, cellular motility, potential cyst formation, invasion, and parasitic pathogenicity [58,61,62,63,64]. EhHSP100 is involved in amoeba response to heat and oxidative stress, protection against cell damage and modulation of pathogenicity [59]. Furthermore, EhMLBP is a mRNA-binding protein known to interact specifically with the 3′ untranslated region of mRNAs of various genes of the parasite, suggesting its potential role in post-transcriptional regulation and messenger RNA stability [60,65,66,67].

Recently, Dorantes et al. [28] conducted an in silico analysis of HSEs located in the promoter regions (ranging from −500 to +50 bp) of the 8343 genes present in the *E. histolytica* genome. This analysis revealed the presence of 2578 HSEs in total, of which 1412 HSEs were in the promoter regions of 1010 hypothetical genes, and 1166 HSEs in the promoter regions of 957 coding genes. Remarkably, 24% of the genes could potentially be regulated by HSEs. Furthermore, it was observed that these HSEs are situated in promoters of genes associated with various functions, including ATP-dependent activity, binding, catalytic activity, cytoskeletal motor activity, molecular adaptor activity, molecular function regulator, molecular transducer activity, structural molecule activity, transcription regulator activity, translation regulator activity, and transporter activity, among others.

These findings further underscore the functional importance of the EhHSTF5 factor as a potential regulator of various genes that harbor HSEs in their promoters in *E. histolytica*. This ability could allow the parasite to dynamically adapt to its environment, thereby ensuring its survival across diverse physiological conditions. Additionally, our study proposes a highly conserved mechanism of EhHSTF5 that transcends *H. sapiens*, *K. lactis*, and *S. cerevisiae*, potentially extending to other parasites where HSTFs have not yet been extensively explored, including *Plasmodium* spp., *Trypanosoma* spp., and *Leishmania* spp. This discovery not only expands our knowledge of parasitic biology and gene expression regulation, but also paves the way for research into new therapeutic targets in these pathogens. This opens up the possibility of developing innovative antiparasitic strategies focused on the activation and regulation of genes by HSTFs.

Current in vitro experiments are underway to evaluate the significance of the “GAA” and “TTC” motifs. Our approach involves targeted manipulation and mutation of these nucleotides within the HSE_*EhPgp5* sequence to understand their impact on binding affinity with rEhHSTF5. Furthermore, conducting assays on *E. histolytica* trophozoites will deepen our comprehension of how oligomerization affects the affinity of EhHSTF5 for various HSEs, particularly under stress conditions. Simultaneously, we will proceed to evaluate the relevance of EhHSTF5 in the parasite by specifically blocking its expression through siRNAs.

In summary, as illustrated in Figure 9, our findings indicate that the mrEhHSTF5 protein, with a molecular weight of 38 kDa, can form dimeric (76 kDa) and trimeric (114 kDa) structures. This oligomerization is crucial, as it increases the stability of the protein, which translates into a higher affinity for HSE_*EhPgp5*.

## 4. Materials and Methods

### 4.1. Cloning of the Ehhstf5 Gene and Expression of the rEhHSTF5 Protein

The *Ehhstf5* gene was cloned into the pET-28a(+) plasmid by GenScript (Piscataway, NJ, USA). The gene was flanked by the restriction sites of the BamHI and XhoI enzymes at the 5′ and 3′ ends, respectively. Additionally, a 6-histidine tag was incorporated at the 5′ end of the plasmid. The insertion of the *Ehhstf5* gene was confirmed through gene amplification. The forward primer (5′-ATCCGATCGGATGAGTGAACCACACAAACA-3′) and reverse primer (5′-CGACTCGACTTTAAAACTTCCATGGAATTT-3′) oligonucleotides were employed. The amplification process started with an initial denaturation step at 94 °C for 5 min, followed by 30 cycles. Each cycle involved denaturation at 94 °C for 1 min, primer annealing at 49.4 °C for 1 min and 30 s, and the extension of the new strand at 72 °C for 1 min. Following the cycles, there was an incubation at 72 °C for 7 min to conclude the amplification process. As a negative control, we conducted the assay without using the plasmid. The resulting PCR product was sequenced using the ABI PRISM 3130 sequencer (Thermo Fisher Scientific, Waltham, MA, USA). Additionally, enzymatic restriction assays were conducted using the pET-28a(+)-*Ehhstf5* plasmid. Using XhoI (New England Biolabs, Ipswich, MA, USA) or BamHI (New England Biolabs, Ipswich, MA, USA) enzymes separately, and the simultaneous application of both enzymes for double restriction. As a negative control, the unrestricted plasmid was used.

The constructed plasmid pET-28a(+)-*Ehhstf5* was used to transform competent cells of *E. coli* C41 strain (Sigma-Aldrich, San Luis, MO, USA). Expression of the recombinant rEhHSTF5 protein was induced using 1 mM Isopropyl-beta-D-thiogalactopyranoside (IPTG) (Thermo Fisher Scientific, Waltham, MA, USA) with agitation at 220 rpm in SOC medium for 24 h at 37 °C. The progress of recombinant protein induction was observed at 3, 6, 9, and 24 h, respectively. As a negative control, bacteria with the plasmid without IPTG were used. Resulting bacterial cells were resuspended in lysis buffer (20 mM Tris-HCl, pH 8, 1 mg/mL lysozyme, and 1% Triton X-100) and then lysed at 4 °C by sonication using an Ultrasonic Processor equipment (MRC, laboratory instruments, Harlow, ESX, UK) at 60% amplitude. Five cycles of 15 s were applied with 15 s rest intervals. The obtained soluble proteins were quantified using the Bradford method [68]. Integrity and expression of the rEhHSTF5 protein were assessed through 12% sodium dodecyl sulfate polyacrylamide gel electrophoresis (SDS-PAGE) using 20 μg of protein.

### 4.2. Immunodetection of rEhHSTF5 Protein by Western Blotting

The proteins separated by electrophoresis were transferred to PVDF membranes with 0.45 μm pores (Sigma-Aldrich, Basel, Switzerland). The membranes were incubated for 30 min with the primary antibody anti-6His (α6xHis) (Sigma-Aldrich, Basel, Switzerland) at a dilution of 1:3000, or with a polyclonal antibody anti-EhHSTF5 (αEhHSTF5) at a dilution of 1:1000, generated in Balb/c mice by SPIDDSNNVELP (GLBiochem, Shanghai, China) EhHSTF5 peptide immunization. As a negative control, serum from non-immunized mice was used, and as a positive control, an αGAPDH antibody (Santa Cruz Biotechnology, Dallas, TX, USA) was employed. The membranes were subsequently incubated for 30 min with the secondary antibody goat anti-mouse IgG (H+L) conjugated to horseradish peroxidase (Thermo Fisher Scientific, Waltham, MA, USA) at a dilution of 1:20,000. Detection was performed through chemiluminescence using Immobilon™ Western chemiluminescent HRP substrate (Millipore, Burlington, MA, USA).

### 4.3. Purification of Recombinant Protein rEhHSTF5

Bacteria containing the overexpressed rEhHSTF5 protein were suspended in buffer I (20 mM Tris-HCl, 500 mM NaCl, and 25 mM imidazole, pH 8.5) and lysed by sonication. The soluble fraction was obtained through centrifugation at 20,000× *g* for 45 min at 4 °C, followed by filtration using 0.22 μm pore size filters. The purification of the recombinant protein was conducted through nickel affinity chromatography using the ÄKTA pure™ chromatography system (Cytiva, Marlborough, MA, USA). Briefly, the protein lysate was loaded onto a 5 mL HisTrap FF column (Cytiva, Marlborough, MA, USA), which was subsequently washed with 5 column volumes (25 mL) of buffer I, and the protein was eluted from the column with buffer II (20 mM Tris-HCl, 500 mM NaCl, 500 mM imidazole, pH 8.5). The equipment facilitated the mixing of both buffers, a capability we utilized to implement four steps with increasing concentrations of imidazole. In the first step, we employed a concentration of 20 mM of imidazole to elute proteins that do not bind or exhibit low affinity for the column. This step was followed by a second step, adjusting the imidazole concentration between 50 and 100 mM. The third step was set in a range of 200 to 300 mM, and finally, the fourth step was fixed at 500 mM to facilitate the complete elution of all proteins from the column. The purification of the rEhHSTF5 protein was evaluated by Western blotting following the methodology described previously.

### 4.4. Oligomerization States of the mrEhHSTF5 Protein

The oligomerization of the monomeric rEhHSTF5 protein (mrEhHSTF5) was assessed by incubating 20 µg of purified protein at 25 °C in the presence of glutaraldehyde (Sigma-Aldrich, St. Louis, MO, USA) at concentrations of 0.5%, 1.5%, and 2.5% for 15 min. As a negative control, we used the mrEhHSTF5 protein without glutaraldehyde. The oligomerization reaction was stopped by adding 10 µL of a loading buffer (2X) containing 100 mM Tris-Cl at pH 6.8, 4% SDS (sodium dodecyl sulfate; electrophoresis-grade), 0.2% bromophenol blue, 20% glycerol, 200 mM DTT (dithiothreitol), and 200 mM β-mercaptoethanol. Oligomerization states were evaluated through 12% SDS-PAGE gel electrophoresis.

### 4.5. Biotinylation and Hybridization of the HSE_EhPgp5 Sequence

The sense (5′-ATAGAAATTTTTCATA-3′) and anti-sense (3′-TATCTTTAAAAAGTAT-5′) oligonucleotides of the HSE_*EhPgp5* were synthesized at Integrated DNA Technologies (IDT) (Coralville, IA, USA). The 3′ end of the single-stranded oligonucleotides was biotinylated using the Biotin 3′ End DNA Labeling Kit (Thermo Fisher Scientific, Waltham, MA, USA), following the manufacturer’s instructions. Subsequently, the oligonucleotides were hybridized in equimolar ratios in an Axygen-Maxigene thermal cycler (Thermo Fisher Scientific, Waltham, MA, USA) at 95 °C for 5 min, followed by sequential descending incubations from 10 °C to 25 °C, holding each temperature for one min.

### 4.6. Electrophoretic Mobility Shift Assays (EMSA)

The interaction between the rEhHSTF5 factor and HSE_*EhPgp5* was evaluated using 0.7 μM of monomeric protein (mrEhHSTF5) or dimeric protein (drEhHSTF5). The proteins were incubated for 30 min at room temperature with the components of the Lightshift EMSA kit (Thermo Fisher Scientific, Waltham, MA, USA), including 2 µL of binding buffer, 1 µL of Poly (dI-dC) (50 ng/µL), 1 µL of glycerol (50%), 1 µL of MgCl_2_ (100 mM), and 300 µg of BSA. For competition assays, a non-biotinylated probe (specific competitor) or Poly (dI-dC) was used as an unspecific competitor at molar excesses of 150 and 350 times, respectively. Prior to incubation with the binding mixture and the biotinylated probe, competitors were preincubated for 30 min at 4 °C with mrEhHSTF5 and drEhHSTF5. As a negative control, only the biotinylated probe without protein was used. The samples were subjected to electrophoresis under native conditions (non-denaturing) on 0.5X-TBE polyacrylamide gels and then transferred to PVDF membranes (Sigma-Aldrich, Basel, Switzerland). The formation of the rEhHSTF5-HSE complex was detected through chemiluminescence using the provided kit solution.

### 4.7. Electrophoretic Mobility Supershift Assay

The αEhHSTF5 antibody was preincubated at a 1:3000 dilution with mrEhHSTF5 or drEhHSTF5 protein for 30 min. This was followed by another 30 min incubation with the binding mixture and biotinylated probe under the same conditions. As negative controls, a pre-immune serum (serum from mice not immunized with the immunogenic peptide of EhHSTF5), an unrelated commercial antibody (αGAPDH, 1:3000), and solely the biotinylated probe without protein were used. The procedure for detecting complexes was carried out following the EMSA conditions described previously. Furthermore, a kinetics study involving an increase in drEhHSTF5 concentration (from 0.7 to 2.8 µM) was conducted to confirm the consistency and correspondence of the formed complexes with the drEhHSTF5-HSE_*EhPgp5* complex.

### 4.8. Three-Dimensional Modeling of the HSE_EhPgp5

We utilized UCSF Chimera version 1.15 software (San Francisco, CA, USA) to construct the three-dimensional model of HSE_*EhPgp5*, using the sense sequence 5′-ATAGAAATTTTTCATA-3′ and anti-sense sequence 3′-TATCTTTAAAAAGTAT-5′. The resulting double-stranded structure in conformation b was saved in .pdb format for subsequent bioinformatics analysis.

### 4.9. 3D Modeling, Structural Validation, and Stability Analysis of mEhDBD5 and dEhDBD5

The amino acid sequence 46-137, corresponding to the DNA-binding domain (DBD) of the EhHSTF5 factor (EhDBD5), was obtained from the AmoebaDB platform (https://amoebadb.org/amoeba/app, accessed on 2 November 2023) with the ID EHI_137000 to generate the 3D models of the DNA-binding domain in monomeric (mEhDBD5) and dimeric (dEhDBD5) conformations. The SWISS-MODEL server [69] (https://swissmodel.expasy.org/, accessed on 2 November 2023) was utilized to model the 3D structures of mEhDBD5 and dEhDBD5, using the DBD of human HSTF2 in complex with HSE as a template, deposited with the ID 5D8L in the Protein Data Bank (PDB) (https://www.rcsb.org, accessed on 2 November 2023). The obtained models were structurally validated using various servers, including PDBsum for Ramachandran analysis (http://www.ebi.ac.uk/thornton-srv/databases/cgi-bin/pdbsum/GetPage.pl?pdbcode=index.html, accessed on 5 November 2023), ProSA-web (https://prosa.services.came.sbg.ac.at/prosa.php, accessed on 10 November 2023), ERRAT and VERIFY 3D contained in UCLA-DOE LAB-SAVES v6.0 (https://saves.mbi.ucla.edu, accessed on 8 December 2023), and VADAR (http://vadar.wishartlab.com/index.html?, accessed on 10 December 2023). Likewise, PDBsum provided secondary structure predictions complementary to the Ramachandran analysis, further improving consistency in the validation of the 3D model in relation to its primary and secondary structures. Additionally, the structural stability of mEhDBD5 was predicted using the normalized B-factor or temperature factor calculated by the I-Tasser server (https://zhanggroup.org/I-TASSER/, accessed on 15 December 2023).

### 4.10. In Silico Molecular Docking

Blind molecular docking simulations were conducted using the HDOCK server (http://hdock.phys.hust.edu.cn/, accessed on 20 December 2023) to assess the intermolecular interactions between HSE_*EhPgp5* and the mEhDBD5 and dEhDBD5 domains of the EhHSTF5 factor. During this process, the .pdb file corresponding to HSE_*EhPgp5* was used as the ligand, while the mEhDBD5 and dEhDBD5 domains were used as receptors. The selection of dockings was based on the most negative values obtained in the Docking Score, representing binding affinity. Additionally, selections were made based on the Confidence score.

### 4.11. Analysis of the Physicochemical Properties of the mEhDBD5-HSE_EhPgp5 and dEhDBD5-HSE_EhPgp5 Complexes

The obtained complexes, mEhDBD5-HSE_*EhPgp5* and dEhDBD5-HSE_*EhPgp5*, were subjected to a comprehensive analysis of physicochemical properties. This analysis covered Aromatics, H-bonds, Charge, Hydrophobicity, Ionizability, and Solvent Accessible Surface (SAS) and was conducted using BIOVIA Discovery Studio, version v.21 (San Diego, CA, USA).

### 4.12. In Silico Analysis of Intermolecular Interactions in the EhDBD5-HSE_EhPgp5 Complex

To identify the intermolecular interactions within the mEhDBD5-HSE_*EhPgp5* and dEhDBD5-HSE_*EhPgp5* complexes, the Protein–Ligand Interaction Profiler (PLIP) server (https://plip-tool.biotec.tu-dresden.de/plip-web/plip/index, accessed on 29 December 2023) was used. Interactions were assessed with a bond length cutoff of 4 Å, following the recommendations of the server.

### 4.13. Structural Similarity of 3D Proteins

The RaptorX server, version 2015 (http://raptorx6.uchicago.edu/, accessed on 2 January 2024) was used to perform 3D structure superposition and obtain structural similarity and root mean square error (RMSD). Among the structures deposited on the PDB platform used for the superposition, the DBD of HSTF in complex with an HSE was used, including that of *Kluyveromyces lactis* (*K. lactis*) (ID; 3HTS); *Saccharomyces cerevisiae*-*Chaetomium thermophilum* (*S. cerevisiae*-*C. thermophilum*, chimera) (ID; 5D5X); *Homo sapiens*-*Chaetomium thermophilum* (*H. sapiens-C*. *thermophilum*, chimera) (ID; 5D5W); *H. sapiens* (ID; 7DCU); and the synthetic *H. sapiens* construct (ID; 5D8L). We were also able to obtain only DBD α2-helix and α3-helix in .pdb format from these structures, which were used to calculate the percentage of structural similarity.

### 4.14. Conservation of DBD α2-Helix and α3-Helix

To evaluate the conservation degree of α-helices 2 and 3 within the DBDs of HS**T**Fs, an amino acid sequence analysis was conducted using the Clustal Omega platform (https://www.ebi.ac.uk/jdispatcher/msa/clustalo, accessed on 6 November 2023). This analysis included the sequences of the seven EhHSTFs of *E. histolytica* previously mentioned, as well as sequences obtained from the National Center for Biotechnology Information platform (NCBI) (https://www.ncbi.nlm.nih.gov/, accessed on 6 November 2023) of HSTF1, HSTF2 and HSTF4 from *H. sapiens* (HsHSTF1, 2 and 4, IDs: 3297, 3298 and 3299), as well as sequences from *Mus musculus* (MmHSTF1, ID: 15499); *S. cerevisiae* (ScHSTF1, ID: 1322586); *K. lactis* (KlHSTF1, ID: 1751387658); *Drosophila melanogaster* (DmHSTF1, ID: 37068); *Caenorhabditis elegans* (CeHSTF1, ID: 45643032); *Gallus gallus* (GgHSTF1, ID: 76523668686); *Arabidopsis thaliana* (AtHSTF1, ID: 827496); *Danio rerio* (DrHSTF1, ID: 134026298); *Xenopus laevis* (XlHSTF1, ID: 148222464); *Pteropus vampyrus* (PvHSTF1, ID: 759162997); *Sus scrofa* (SsHSTF1, ID: 100511321); *Brugia malayi* (BmHSTF1, ID: CRZ24125. 1); *Dasypus novemcinctus* (DnHSTF1, ID: 101424131); *Cavia porcellus* (CpHSTF1, ID: 101787582); *Phascolarctos cinereus* (PcHSTF1, ID: 1190400766); *Pan troglodytes* (PtHSTF1, ID: 741396); *Bos taurus* (BtHSTF1, ID: 506235); and *Capra hircus* (ChHSTF1, ID: 102178552).

Simultaneously, amino acid sequences were extracted from the crystals deposited in the PDB database for *K. lactis* (ID: 3HTS); *S. cerevisiae-C. thermophilum* (ID: 5D5X); *H. sapiens-C. thermophilum* (ID: 5D5W); synthetic *H. sapiens* construct (ID: 5D8L); and *H. sapiens* (ID: 7DCU) to perform a sequence alignment and determine the percentage of identity and homology.

Complementarily, logos were constructed using the amino acids from the aforementioned sequences through the WebLogo server, Version 2.8.2 (https://weblogo.berkeley.edu/logo.cgi, accessed on 7 November 2023).

### 4.15. 3D Structure Viewer

PyMol, version 2.5.0, (Nueva York, NY, USA) and BIOVIA Discovery Studio were used to visualize and generate images for different bioinformatics analyses.

### 4.16. Relative Expression and Graphics

The ImageJ software, version 1.54, (Bethesda, MD, USA) was used to quantify the relative protein expression through pixel analysis. For the statistical analysis, we assessed the normality of the data distribution using the Shapiro–Wilk test, chosen for its high sensitivity with small to moderate sample sizes [70]. This evaluation was conducted using GraphPad Prism software, version 8.0.1. Upon confirming the normal distribution of the data, in accordance with the normality criterion (*p* > 0.05), we subsequently performed a two-way ANOVA, following the American Psychological Association (APA) guidelines for adjusted *p*-value analysis. Statistical significance was categorized as follows: * *p* ≤ 0.033, ** *p* ≤ 0.002, and *** *p* ≤ 0.001. The results are presented as mean ± standard deviation, derived from a minimum of three independent experiments. Following the ANOVA analysis, we applied Tukey’s multiple comparison test to identify significant differences between study groups. For graph creation, the GraphPad Prism software, version 8.0.1 (Boston, MA, USA), was used, allowing for a precise and detailed graphical representation of the results.

## 5. Conclusions

The oligomerization of the rEhHSTF5 factor is a crucial process that enhances its affinity for recognizing the HSE of the *E. histolytica EhPgp5* gene. The specific ability of drEhHSTF5 to identify the structural motifs “GAA” and “TTC” within the HSE suggests its potential role in regulating the expression not only of this gene, but also of others that harbor HSEs in their promoters and that participate in fundamental processes for the parasite, including development, growth, metabolism, infection, virulence, encystment, transition from cyst to trophozoite and response to different types of stress, among others. Therefore, EhHSTF5 may play a fundamental role in the adaptation of the parasite to its environment and contribute to its survival.

## Figures and Tables

**Figure 1 ijms-25-04218-f001:**
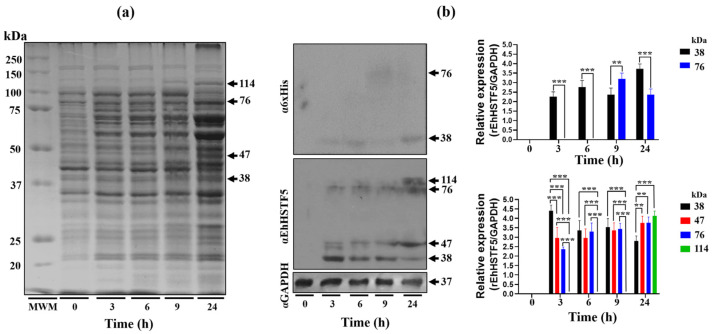
Induction and immunodetection of the oligomeric states of the rEhHSTF5 protein. The integrity of the total proteins extracted from transformed and IPTG-induced bacteria was evaluated by SDS-PAGE. A Molecular Weight Marker (MWM) and IPTG induction times ranging from 0 to 24 h were included. The gel was stained with Coomassie Brilliant Blue G-250 to visualize the protein bands (**a**). Subsequently, immunodetection was performed by Western blot assays using α6xHis and αEhHSTF5 specific antibodies to detect the distinct conformations of rEhHSTF5 (**b**). The graphs display the relative protein expression, calculated as the average of the pixel values obtained from three independent samples. Expression was normalized using the positive control, αGAPDH. The statistical analysis was performed using a two-way ANOVA, followed by the Tukey’s multiple comparison test. Significant differences between experimental groups are denoted with (*). The significance levels used were ** *p* ≤ 0.002, and *** *p* ≤ 0.001.

**Figure 2 ijms-25-04218-f002:**
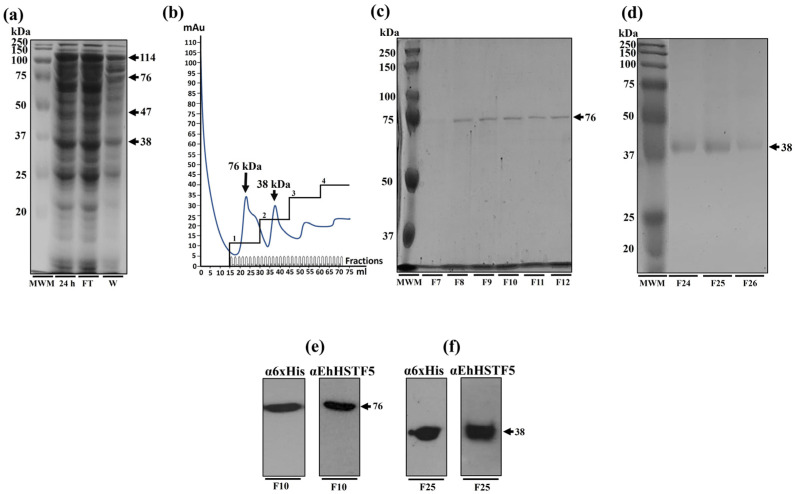
Purification and immunodetection of oligomeric conformations of the rEhHSTF5 protein. The oligomeric conformations of the rEhHSTF5 protein were monitored before and after loading onto the HisTrap FF column. The total proteins obtained from the bacteria, transformed with the plasmid construct and induced with IPTG (24 h), were passed through filters with a pore size of 0.22 μm (flowthrough, FT), and the proteins that did not bind to the column (waste, W) are shown (**a**). The chromatogram and SDS-PAGE assays revealed that the drEhHSTF5 protein was eluted in fractions 7 to 12, while the mrEhHSTF5 protein was eluted in fractions 24 to 26 (**b**–**d**). These proteins were detected using α6xHis and αEhHSTF5 antibodies in WB assays (**e**,**f**). In the chromatogram (**b**), the four increasing imidazole steps are illustrated: the first corresponds to a concentration of 20 mM, followed by the second with 50–100 mM, the third between 200–300 mM, and the fourth at 500 mM, while the blue line indicates the absorbance reading at 280 nm during the elution time, signifying the presence of proteins eluted from the affinity column. For Panels (**c**–**f**), “F” denotes the fraction number obtained from the chromatography, and “MWM” indicates the molecular weight marker.

**Figure 3 ijms-25-04218-f003:**
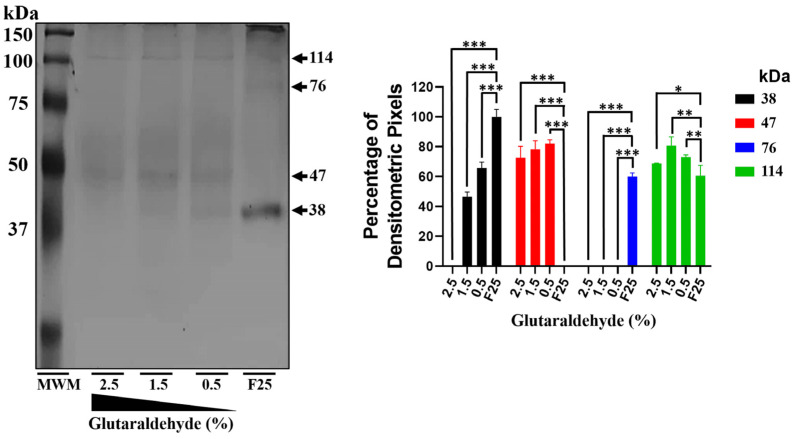
mrEhHSTF5 oligomerization induction with increasing concentrations of glutaraldehyde. We used the mrEhHSTF5 protein without glutaraldehyde as a negative control. Oligomerization states were evaluated by 12% SDS-PAGE gel electrophoresis stained with Coomassie Brilliant Blue G-250. The statistical analysis was performed using a two-way ANOVA, followed by the Tukey’s multiple comparison test. Significant differences between experimental groups are denoted with (*). The significance levels used were * *p* ≤ 0.033, ** *p* ≤ 0.002, and *** *p* ≤ 0.001.

**Figure 4 ijms-25-04218-f004:**
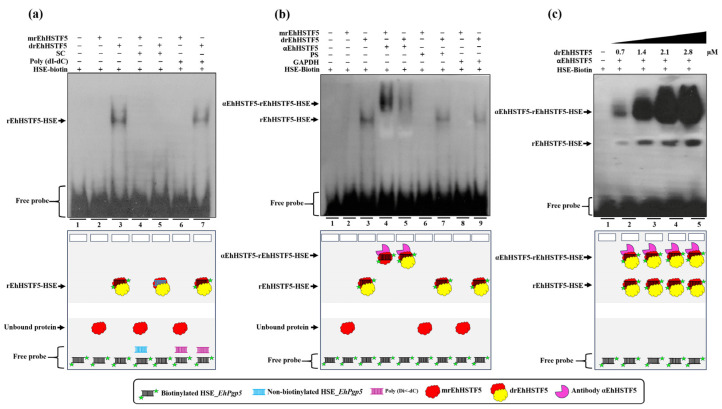
HSE recognition of the *EhPgp5* gene by the rEhHSTF5 protein. EMSA assays were performed using the monomeric and dimeric conformations of the rEhHSTF5 protein. A specific competitor (non-biotinylated HSE_*EhPgp5*) and a nonspecific competitor (Poly(dI-dC)) were used (**a**). For the supershift assays, the specific antibody αEhHSTF5 was utilized for both conformations of the rEhHSTF5 protein. Unrelated antibodies, including mouse preimmune serum (PS) and the αGAPDH antibody, were also used as negative controls (**b**). Likewise, a kinetic study with increasing drEhHSTF5 concentration (from 0.7 to 2.8 µM) drEhHSTF5 was performed in supershift assays (**c**). As a negative control, only the biotinylated probe without protein was used. In the image legend, the plus sign (+) indicates the presence of a component in the reaction, while the minus sign (−) denotes its absence. A graphical illustration of each condition used in the EMSA and supershift tests is provided at the bottom.

**Figure 5 ijms-25-04218-f005:**
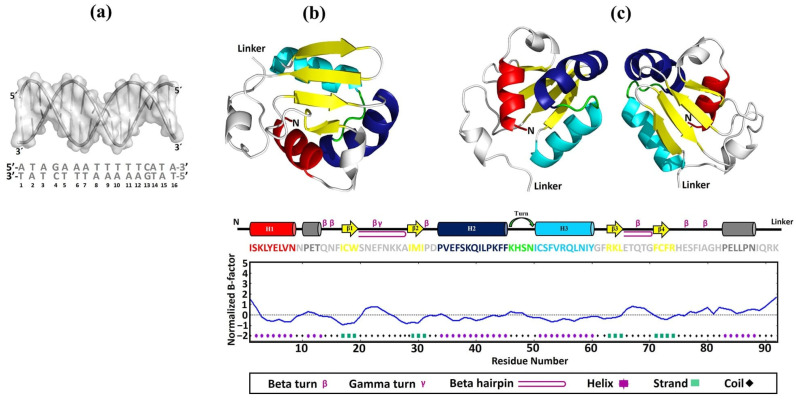
Construction of 3D models. The 3D structures of HSE_*EhPgp5* (**a**), mEhDBD5 (**b**), and dEhDBD5 (**c**) were obtained. Additionally, the representation DBD in the factor EhHSTF5 is illustrated, covering everything from the amino acid sequence to its secondary and three-dimensional structures. The representation includes both the monomeric conformation and the potential homodimeric conformation of the domain. The primary and secondary structures of EhDBD5 are displayed using a color code to facilitate understanding, while the corresponding graph illustrates its stability, assessed in silico through the normalized B-factor provided by the I-TASSER server.

**Figure 6 ijms-25-04218-f006:**
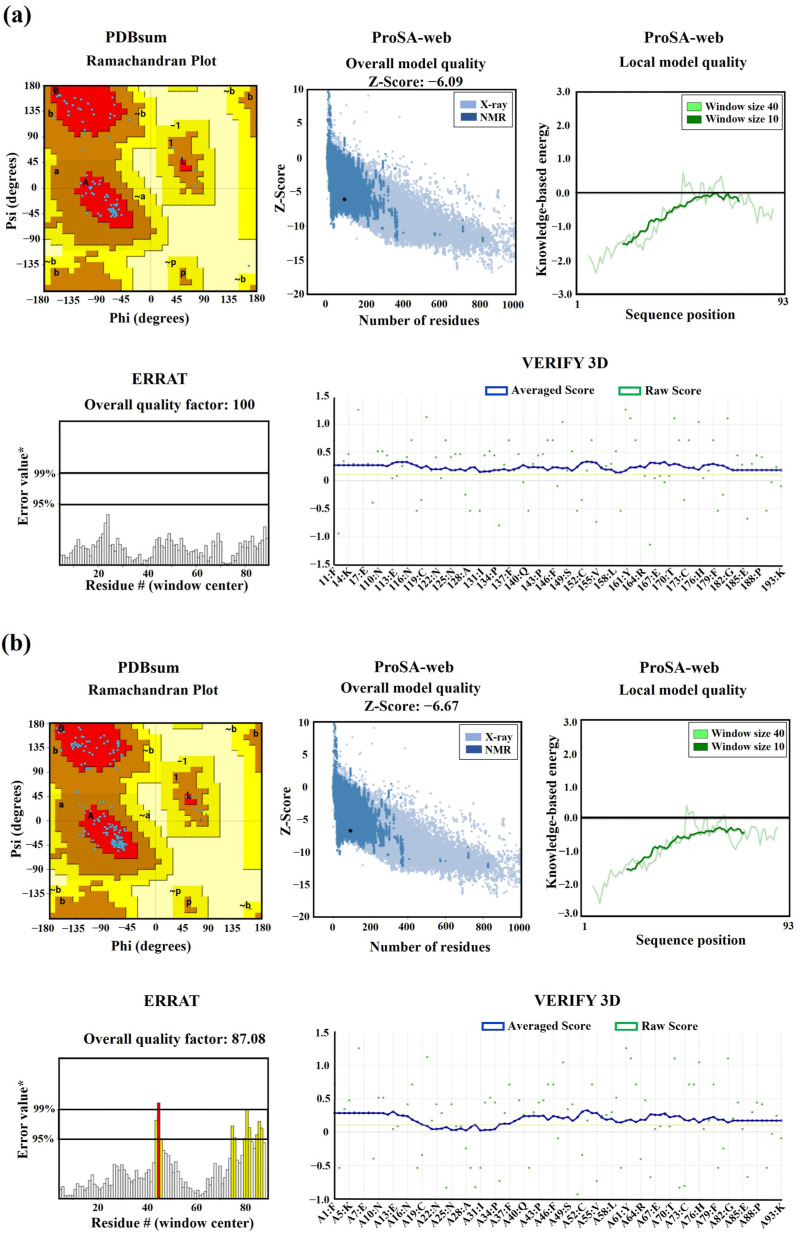
Validation of 3D structures. The 3D models mEhDBD5 (**a**) and dEhDBD5 (**b**) were validated using Ramachandran analysis, which provides detailed information about the distribution of dihedral angles of amino acids in favorable and unfavorable regions. The capital letters A, B, and L are used to designate the Most Favored Regions, while the lowercase letters a, b, l, and p represent the Additional Allowed Regions. The tilde (~a, ~b, ~l, ~p) is utilized to indicate the Generously Allowed Regions. Furthermore, a value of −1 is assigned to a torsion angle located within a non-permissible region. The models were subjected to the ProSA-web server for a validation based on the Z-score (the black dot on the graph represents the Z-score assigned to our 3D structure) and global energy, where values ≤ 0 indicate absence of structural errors. ERRAT was used for error quantification, where an asterisk (*) indicates the error value. White bars in the graph represent instances without structural errors, while yellow bars indicate amino acids with structural errors falling within the 95–99% error range, and red bars identify errors exceeding 99%. Verifying 3D structures validates protein structures by evaluating their 3D conformation through comparison to a set of experimental structures. It assigns a score, suggesting an acceptable average 3D−1D score of ≥0.1 for validation.

**Figure 7 ijms-25-04218-f007:**
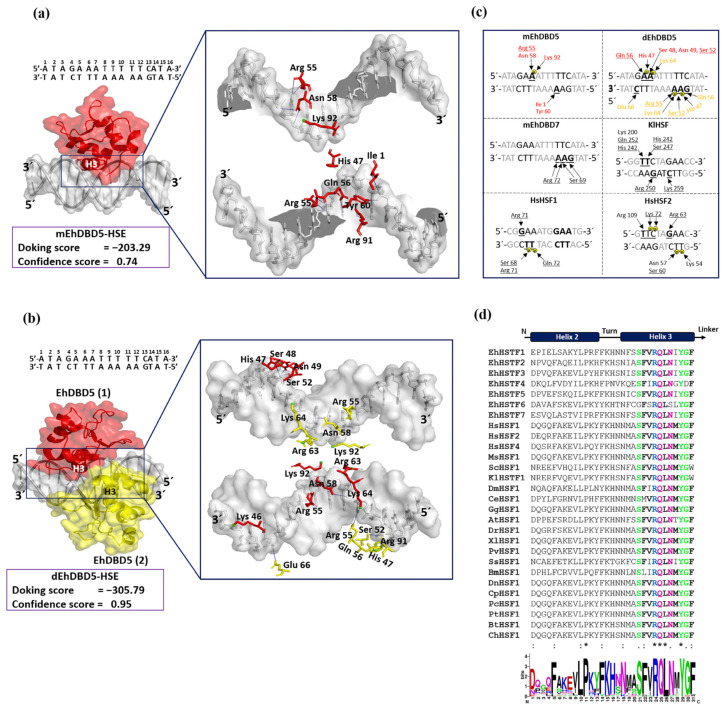
Molecular docking and intermolecular evaluation of the mEhDBD5-HSE_*EhPgp5* and dEhDBD5-HSE_*EhPgp5* complexes. Blind dockings were performed with the domains mEhDBD5 or dEhDBD5 and the HSE_*EhPgp5*. Using the PLIP server, intermolecular interactions of amino acids involved in the interaction with the sense and anti-sense strands of HSE were estimated. A close-up of the area of interaction is shown in the boxes, where amino acids from mEhDBD5 (EhDBD5(1)) are illustrated in red, while the analogue of the second EhDBD5 domain from dEhDBD5 (EhDBD5(2)) is shown in yellow (**a**,**b**). The amino acids involved in the interaction with the amino acids of the mEhDBD7-HSE, KlHSTF-HSE, HsHSTF1-HSE, and HsHSTF2-HSE complexes were compared (**c**). The high conservation degree of α2-helix and α3-helix of DBDs from 28 species, including the EhHSTFs family, is shown (**d**). In the alignment with the Clustal Omega server, asterisks (*) indicate amino acids that are identical across all sequences. Colons (:) represent conserved substitutions with similar properties, and a period (.) points out semi-conserved substitutions with some functional similarity. The analysis of the physicochemical properties of the complexes revealed that the dEhDBD5-HSE_*EhPgp5*, containing an additional monomer compared to the mEhDBD5-HSE_*EhPgp5*, demonstrated more significant aromatic interactions, hydrogen bonds, and charges. This complex also exhibited an increased hydrophobicity and a higher ionization capacity. Furthermore, it possessed a more extensive solvent-exposed surface area than its monomeric counterpart, mEhDBD5-HSE_*EhPgp5* (Appendix A).

**Figure 8 ijms-25-04218-f008:**
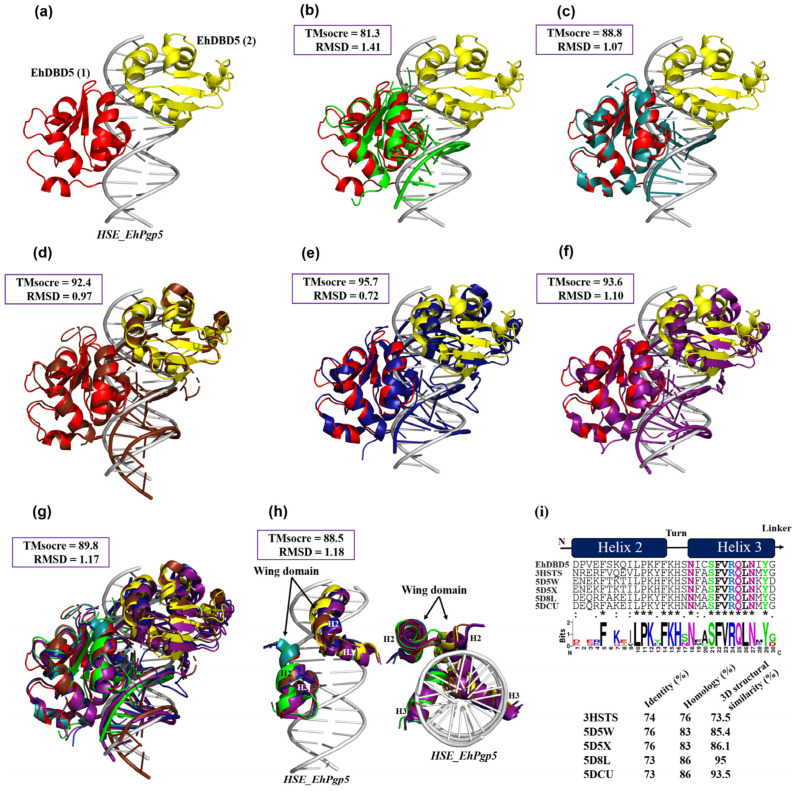
The 3D structural similarity between DBDs–HSEs complexes. Based on the obtained dEhDBD5-*HSE_EhPgp5* complex (**a**), overlay comparisons were performed with crystallographic structures deposited in the PDB database. These included the *K. lactis* DBD-HSE complex (ID: 3HTS) (**b**), *S. cerevisiae*-*C. thermophilum* complex (ID: 5D5X) (**c**), *H. sapiens*-*C. thermophilum* complex (ID: 5D5W) (**d**), synthetic *H. sapiens* construction (ID: 5D8L) (**e**), and *H. sapiens* (ID: 7DCU) (**f**). A joint overlay of all these structures was performed (**g**) and focusing solely on the wing domain (**h**). In addition, a significant degree of conservation was identified through the alignment of amino acid sequences of helices 2 and 3 (**i**). The squares illustrate the TM-scores in percentages, while the RMSD is in Angstroms (Å). In the alignment with the Clustal Omega server, asterisks (*) indicate amino acids that are identical across all sequences. Colons (:) represent conserved substitutions with similar properties, and a period (.) points out semi-conserved substitutions with some functional similarity.

**Figure 9 ijms-25-04218-f009:**
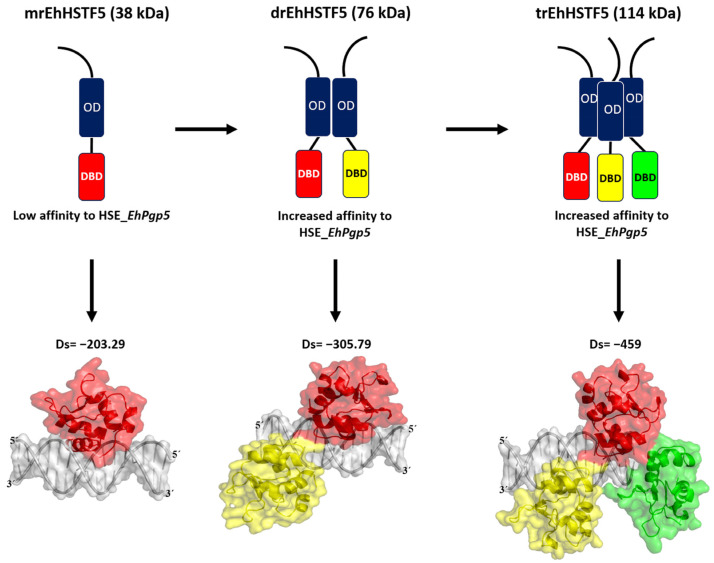
Our in vitro findings demonstrate that the mrEhHSTF5 protein is capable of oligomerizing, forming dimers and trimers. Furthermore, through EMSA assays and in silico molecular docking, we discovered that the dimeric conformation exhibits greater affinity for HSE_*EhPgp5* compared to the monomeric conformation. Dorantes et al. [28] reported higher affinity (−459) binding of tEhHSTF5 to the same HSE, supporting our results and indicating that the degree of oligomerization increases the affinity for HSE_*EhPgp5*. OD; Oligomerization Domain, DBD; DNA Binding Domain, Ds; Docking Score.

## Data Availability

The datasets used and/or analyzed during the current study are available from the corresponding author upon reasonable request.

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
