# Peer review of "Entamoeba histolytica: In Silico and In Vitro Oligomerization of EhHSTF5 Enhances Its Binding to the HSE of the EhPgp5 Gene Promoter"

_ijms, 2024, doi:10.3390/ijms25084218_

Round 1

Reviewer 1 Report

Comments and Suggestions for Authors

The authors use a poly-histidine tagged recombinant protein to study the dimerization of a transcription factor (EhHSTF5) from the human parasite Entamoeba histolytica. The work is of high quality and the manuscript is very well written and clear. However, the use of the 6-histidine tag may produce problems as described by Wu & Filutowicz 1999 in very similar application. This could be resolved by the removal of the tag after purification. This is normally accomplished by the inclusion of a specific protease site (such as the tobacco etch virus protease) after the tag.

Wu, J., & Filutowicz, M. (1999). Hexahistidine (His6)-tag dependent protein dimerization: a cautionary tale. Acta Biochimica Polonica46(3), 591-599.

Figure 2. The diagram in part b shows a stepwise elution but what the composition of the elution buffers in each if these steps?  The method mentioned just one elution buffer that is buffer II (20 mM Tris-HCl, 500 mM NaCl, 733 500 mM Imidazole, pH 8.5).

Figure 3. Glutaraldehyde was used to crosslink the complex, but would it not have been better to use a zero-length crosslinker such as EDC?

Why did the authors not use gel filtration to study this complex?  This would seem to be the obvious technique to employ. Another would be native gel electrophoresis?

Reviewer 2 Report

Comments and Suggestions for Authors

Objective of this study was the investigation of the potential oligomerization and the DNA-binding affinity of the recombinant EhHSTF5 protein (rEhHSTF5) to the HSE_EhPgp5.

Introduction

-The authors should make clear the gap in the literature that their work is trying to fill.

-The novelties of the study must be clearly underlined and commented upon.

Materials and methods

-The authors have used improved methodologies in comparison to previous relevant studies and these must be explained in greater detail. Also, this must be explained in the discussion.

-The authors must add a new subsection to title it ‘Controls’ and in there they must describe clearly the positive and negative controls used in the study.

-The statistical analysis misses the step of determining whether results have a normal distribution, so this must be performed.

Results

-Figures 5 and 8. Please present other perspectives of the protein structures shown in these figures. As these presentations are now, they are not convincing.

Discussion

-Please divide in sub-sections to allow better flow of reading.

-The Discussion does not address all the main issues raised in the study, so please rewrite and include all aspects of the work performed.

-Some recent relevant references are missing.

-Can the authors postulate please to which other eukaryotes is their work applicable?

Overall. The manuscript requires significant improvement before possible acceptance.

Round 2

Reviewer 2 Report

Comments and Suggestions for Authors

The authors have made extensive corrections and have indeed improved the manuscript.

Before acceptance, I suggest to add some of the comments in the reply regarding the presentation of the structures of the protein in Figures 5 and 8 within the main text, as this will enrich the final paper.
